# Fe-Doped g-C_3_N_4_: High-Performance Photocatalysts in Rhodamine B Decomposition

**DOI:** 10.3390/polym12091963

**Published:** 2020-08-30

**Authors:** Minh Nguyen Van, Oanh Le Thi Mai, Chung Pham Do, Hang Lam Thi, Cuong Pham Manh, Hung Nguyen Manh, Duyen Pham Thi, Bich Do Danh

**Affiliations:** 1Center for Nano Science and Technology, Hanoi National University of Education, 136 Xuan Thuy Road, Cau Giay District, Hanoi 100000, Vietnam; Minhsp@gmail.com (M.N.V.); phammanhcuonghd1995@gmail.com (C.P.M.); 2Department of Physics, Hanoi National University of Education, 136 Xuan Thuy Road, Cau Giay District, Hanoi 100000, Vietnam; phamdochung@gmail.com (C.P.D.); dodanhbich@hnue.edu.vn (B.D.D.); 3Faculty of Basic Sciences, Hanoi University of Natural Resources and Environment, 41A Phu Dien Road, North Tu Liem, Hanoi 100000, Vietnam; lamhang289@gmail.com; 4Nguyen Trai Specialized Senior High School, Haiduong 03000, Vietnam; 5Department of Physics, Hanoi University of Mining and Geology, Duc Thang ward, North Tu Liem District, Hanoi 100000, Vietnam; manhhungmdc@gmail.com; 6Military Science Academy, 322 Le Trong Tan street, Dinh Cong, Hoang Mai, Hanoi 100000, Vietnam; linhduyen1987@gmail.com

**Keywords:** Fe-doped g-C_3_N_4_, photocatalytic performance, interstitial, recombination rate

## Abstract

Herein, Fe-doped C_3_N_4_ high-performance photocatalysts, synthesized by a facile and cost effective heat stirring method, were investigated systematically using powder X-ray diffraction (XRD), Fourier transform infrared (FTIR), scanning electron microscopy (SEM) and Brunauer–Emmett–Teller (BET) surface area measurement, X-ray photoelectron (XPS), UV–Vis diffusion reflectance (DRS) and photoluminescence (PL) spectroscopy. The results showed that Fe ions incorporated into a g-C_3_N_4_ nanosheet in both +3 and +2 oxidation states and in interstitial configuration. Absorption edge shifted slightly toward the red light along with an increase of absorbance in the wavelength range of 430–570 nm. Specific surface area increased with the incorporation of Fe into g-C_3_N_4_ lattice, reaching the highest value at the sample doped with 7 mol% Fe (FeCN7). A sharp decrease in PL intensity with increasing Fe content is an indirect evidence showing that electron-hole pair recombination rate decreased. Interestingly, Fe-doped g-C_3_N_4_ nanosheets present a superior photocatalytic activity compared to pure g-C_3_N_4_ in decomposing RhB solution. FeCN7 sample exhibits the highest photocatalytic efficiency, decomposing almost completely RhB 10 ppm solution after 30 min of xenon lamp illumination with a reaction rate approximately ten times greater than that of pure g-C_3_N_4_ nanosheet. This is in an agreement with the BET measurement and photoluminescence result which shows that FeCN7 possesses the largest specific surface area and low electron-hole recombination rate. The mechanism of photocatalytic enhancement is mainly explained through the charge transfer processes related to Fe^2+^/Fe^3+^ impurity in g-C_3_N_4_ crystal lattice.

## 1. Introduction

Humanity and other life forms on the earth are facing serious threats due to the exhaustion of fossil fuel resources together with the problem of environmental pollution caused by human activities in industry and life [1,2]. Therefore, the search for alternative resources of fuel, both abundant and environmentally friendly, is a vital task of humanity [3]. In addition, research on finding new materials to treat hazardous waste from industry and life such as liquid waste, exhaust gas, solid waste is also an essential task [4,5].

The advanced oxidation processes (AOPs) [6,7,8] have been well known to be the effective method in removing toxic organic materials from water, gas and soil, as well as in producing hydrogen fuel gas form water. The characteristic of this process is the photochemical reaction between semiconductors and light (visible or ultraviolet), creating photogenerated electron-hole pairs. These particles oxidize and reduce either organic pollutions directly or in the other hand react with oxygen and water molecules to form oxidative radicals of super oxides (O_2_^•^) and hydroxyl (OH^•^). These chemical species can oxidize and mineralize organic substances into CO_2_ and water or at least promote the conversion of toxic organic contaminations into non-toxic organic substances such as destroying the aromatic ring [9].

Semiconductors that possess attractive properties such as low cost, non-toxic, mechanically and thermally durable, high efficiency of electron-hole pairs production, low electron-hole recombination rate have been targeted as the best and most flexible for AOPs for several decades [10,11,12,13,14]. In particular, several typical compounds can be mentioned as titanium dioxide TiO_2_ [15,16], perovskite materials ABO_3_ [17,18,19,20], zinc oxide ZnO [21,22], zinc tungsten oxide ZnWO_4_ [23,24,25] and so forth. However, a common feature of this semiconductor generation is the large band gap (E_g_~3.2 eV) which restricts the efficiency of sunlight in stimulating photochemical reactions. Recently, graphitic carbon nitride g-C_3_N_4_ has emerged as a non-metallic semiconductor with many outstanding advantages, satisfying the requirements of an efficient photocatalytic semiconductor to generate the large number of electron-hole pairs and the application of sunlight in photochemical reactions due to the narrow optical band gap (E_g_~2.7 eV) [26,27]. In addition, the nanosheet structure of g-C_3_N_4_ with high porosity and large specific surface area also supports for photocatalytic application of this material. However, studies showed that photogenerated electrons in g-C_3_N_4_ are localized in each heptazine unit, resulting to a high recombination rate of electron-hole pairs and preventing photocatalytic activity of material [28].

Studies based on g-C_3_N_4_ have been mostly carried out in attempt to reduce the recombination rate of photogenerated electron-hole pairs, in addition to reduce the energy band gap. Effective procedures may be referred to as—(i) creating heterojunctions like semiconductor-semiconductor [29,30,31] or metal-semiconductor [32,33,34] in order to effectively separate electrons and holes; (ii) doping metallic elements into g-C_3_N_4_ as electron trapping centers [35,36], leaving highly oxidized holes. Recent works showed that the incorporation of metal ions into g-C_3_N_4_ polymerization network not only improves charge carrier lifetime and mobilities but also reduces the band gap of g-C_3_N_4_. In g-C_3_N_4_ network, tri-s-triazine units connected by 3-fold N-bridges create a large space with six lone-pair electron nitrogen atoms, which can serve as an ideal coordination for accommodation of transition metal ion (Figure 1).

Since Wang et al. observed the modify of functionality of g-C_3_N_4_ as doped with some metals including Fe in 2009 [37], Fe-doped g-C_3_N_4_ nanosheets have been studied to find out highly effective photocatalysts as well as explore the mechanism to improve their photocatalytic activity. In 2014, Fe-doped g-C_3_N_4_ nanosheets synthesized from melamine and FeCl_3_ by a two-step method was studied by Tonda et al. [38]. The significant improvement of photocatalytic performance for Rhodamine B degradation was explained by the role of Fe^3+^ as the trap of photogenerated electrons. In 2019, Ma et al. reported on Fe-doped g-C_3_N_4_ photocatalyst synthesized by one-step thermal condensation of iron nitrate nonahydrate (Fe(NO_3_)_3_·9H_2_O) and melamine [35] in which the authors emphasized the pivotal role of Fe^3+^/Fe^2+^ couple in the photocatalytic reaction. In addition, studies on (Fe, P) co-doped g-C_3_N_4_ also shown significant improvement in photocatalytic efficiency [39,40]. Evidence of the improvement of photocatalytic efficiency as well as its mechanism in Fe-doped g-C_3_N_4_ material remains to be confirmed, contributing greatly to the discovery of high-performance photocatalyst in particular and solar energy conversion material in general.

In this study, Fe-doped g-C_3_N_4_ was synthesized from urea and FeCl_3_ by a two-step method. The study will provide the evidence of Fe impurity position in g-C_3_N_4_ host lattice, demonstrate the presence of Fe^3+^/Fe^2+^ couple as well as discuss their role in photocatalytic performance enhancing.

## 2. Materials and Methods

### 2.1. Synthesis of g-C_3_N_4_ Nanosheets

An appropriate amount of urea (NH_2_CONH_2_, >99%, Sigma-Aldrich, China) contained in a sealed glass was heated at 550 °C for 2 h in air atmosphere. The bright yellow g-C_3_N_4_ product [41,42] was dispersed in distilled water, ultrasonicated for 3 h and then dried to obtain g-C_3_N_4_ nanosheets.

### 2.2. Synthesis of Fe-dope g-C_3_N_4_ Nanosheets

To synthesize Fe-doped g-C_3_N_4_, 0.5 g of g-C_3_N_4_ nanosheets was dispersed in 50 mL of distilled water by magnetically stirring for 30 min and then ultrasonicating for 1 h to obtain a suspension of g-C_3_N_4_ nanosheets. An appropriate amount of ferric chloride hexahydrate (FeCl_3_.6H_2_O, >98%, Sigma-Aldrich, China) corresponding to 3, 5, 7, 8, 10 mol% of Fe was added to above solution which was continuously heated for 12 h at temperature of 90 °C under magnetically stirring condition in order to allow Fe atoms to insert into appropriate interstitial positions in g-C_3_N_4_ crystal lattice. The suspension was rinsed with ethanol and centrifuged at 6000 rpm for 10 min for 3 times to remove excess contaminations from the solution. The remaining material was dried at 80 °C for 5 h to achieve Fe-doped g-C_3_N_4_ nanosheets.

For a convenient, g-C_3_N_4_ photocatalysts doped with 3, 5, 6, 7, 8 and 10 mol% of Fe were denoted as FeCN_3_, FeCN_5_, FeCN_6_, FeCN_7_, FeCN_8_ and FeCN_10_, respectively.

### 2.3. Characterizations

X-ray diffraction (XRD) patterns of Fe-doped g-C_3_N_4_ nanosheets were recorded by a D8 Advance diffractometer (Bruker, Germany) using Cu-K_α_ radiation. Lattice parameters were calculated by using UnitCell software. IR Prestige-21 FTIR/NIR spectrometer (Shimadzu, Japan) was used to carry out Fourier transform infrared spectra (FTIR) of as-synthesized samples. Scanning electron spectroscopy (SEM) images of fabricated samples were obtained using a S-4800 NIHE microscope (Hitachi, Japan). The Brunauer–Emmett–Teller (BET) surface area was measured by a 3Flex (Micromeritics, America). UV–Vis diffuse reflectance spectra (DRS) was performed on a V670 UV–Vis spectrophotometer (Jasco, Japan). Multilab-2000 spectrometer with an Al Kα monochromatized source was used to measured X-ray photoelectron spectroscopy (XPS). Photoluminescence (PL) spectra were performed on a Nanolog iHR 320 fluorescence spectrophotometer (Horiba, Japan) using an excitation wavelength of 350 nm.

### 2.4. Photocatalytic Process

Rhodamine B chemical compound was chosen as a degradation agent in determining photocatalytic activity of Fe-doped g-C_3_N_4_ nanosheets. Firstly, a solution of RhB 20 ppm was prepared. Fe-doped g-C_3_N_4_ photocatalyst was weighed with appropriate amount of 0.06 g and dispersed in 30 mL of distilled water DI under ultrasound vibration for 1 h. Due to nanosheet morphology of sample the solution was formed as opaque with tiny particle suspended. Next, dissolving 30 mL of RhB 20 ppm solution into 30 mL of above g-C_3_N_4_ contained solution under magnetically stirring condition. The solution was stirred for 30 min in the dark to reach equilibrium state of adsorption-desorption. To check for equilibrium, 3 mL of solution was removed after each 10 min of dark stirring to determine the remaining concentration of RhB. After that, the solution was stirring under irradiating condition of a 300 W xenon lamp at a distance of 10 cm from lamp to liquid surface (or under sunlight illumination). A piece of glass was used to block ultraviolet irradiation from xenon lamp entering the photocatalytic solution. An amount of 3 mL of solution was removed after each 10 min of exposure and filtered g-C_3_N_4_ nanosheets to measure UV–Vis absorption spectra. The absorbance of 554 nm peak of RhB was used to determine the remaining content of RhB in the solution at a time by using a standard curve that represents the relationship between RhB concentration and absorbance.

## 3. Results and Discussions

### 3.1. Structural Property Analysis

Figure 2a shows XRD patterns of Fe-doped g-C_3_N_4_ with different Fe concentrations which reveals the existence of three XRD peaks at 12.90°, 24.90° and 27.62°. This confirms that as-synthesized samples are graphitic carbon nitride with hexagonal structure (JCPDS card no. 87-1526) in which observed diffraction planes are (001), (101) and (002), respectively. Lattice parameters calculated from the position of XRD peaks are *a* = *b* = 4.96 Å and *c* = 6.46 Å (the interlayer stacking distance equals *c*/2 = 3.23 Å). All samples do not show any trace of Fe species within the detection limit of conventional XRD. In addition, (101) and (002) diffraction peaks slightly shift to the left side as Fe concentration increase (Figure 2b). Lattice parameters were calculated as (a = b = 4.97, c = 6.47) and (a = b = 4.98, c = 6.48) for FeCN3 and FeCN5 photocatalysts, respectively. The increase in lattice parameters reveals a certain lattice disorder of g-C_3_N_4_ as doping Fe, leading to a less dense packing fashion in crystal lattice. This change can be attributed to the interstitial doping configuration of large radius Fe ions in g-C_3_N_4_ by chemically coordinating with six lone-pair electron nitrogen atoms as shown in Figure 1, resulting in the lattice expansion.

Figure 3a displays Fourier transform infrared (FTIR) absorption spectra of pure g-C_3_N_4_ and Fe-doped g-C_3_N_4_ with different Fe concentrations. The broad absorption band at 3170 cm^−1^ could be assigned to the stretching vibrational modes of residual N–H components associated with uncompensated amino groups. The peak at 1638 cm^−1^ is indexed for stretching vibrational modes while bands at 1570, 1406, 1320 and 1240 cm^−1^ are associated with aromatic C–N stretching vibrations. The sharp characteristic peak at 814 cm^−1^ corresponds to the breathing mode of the s-triazine ring. The intensity of all peaks increases with increasing Fe content. The magnification of FTIR absorption (Figure 3b) peaks indicates the slight shift of 814 cm^−1^ peak towards the higher wavenumber as Fe content increases to 812.1, 813.4 and 813.9 cm^−1^ for g-C_3_N_4_, FeCN5 and FeCN7, respectively while the peaks at 1240 cm^−1^ and 1320 cm^−1^ almost do not change the position. This continues to reveal the influence of Fe-doping on g-C_3_N_4_ crystal lattice, albeit very small, leading to the slight expansion of the benzene ring as observed in XRD analysis.

### 3.2. Morphology Analysis

Field emission scanning electron microscopy (FE-SEM) images of pure g-C_3_N_4_ sample are shown in Figure 4 which reveal the nanosheet morphology of as-synthesized g-C_3_N_4_ in air atmosphere. The nanosheets are very thin, about of 10 nm in thickness, ranging in width from several tens to several hundreds of nanometers. SEM images show that the sample has a high porosity, the nano sheets have a certain curvature, stacked on top of each other, creating the slit-shaped pores with several tens of nanometer in diameter which is convenient for contacting between photocatalyst and organic molecules in photocatalytic process.

Specific surface area and pore size distributions of pure g-C_3_N_4_, FeCN5, FeCN7 and FeCN10 photocatalysts were measured by nitrogen adsorption isotherm analysis (N_2_-absorbtion/desorption) and presented in Figure 5 and Table 1. Figure 5a shows that all samples have IV-type isotherm with H3 hysteresis loop of a mesoporous structure where slit shaped pores are created from non-uniform size and/or shape plates as g-C_3_N_4_ nanosheets [43]. High-pressure hysteresis loop with P/P_o_ > 0.9 responses for macropores. The BET surface area is 91 m^2^/g, 100 m^2^/g, 132 m^2^/g and 104 m^2^/g for g-C_3_N_4_, FeCN5, FeCN7 and FeCN10, respectively, which indicates that specific surface area increases with the incorporation of Fe into g-C_3_N_4_ lattice. Since large specific surface area is benefit for photocatalytic activity, FeCN7 with largest BET surface area will be expected to have a high photocatalytic performance. All samples exhibit a wide range of pore size distribution (Figure 5b) and a large average pore width in the range of 35–40 nm.

### 3.3. Optical Property Analysis

UV–Vis diffusion reflectance spectra (DRS) was used to evaluate the influence of Fe corporation on optical property and energy structure of host g-C_3_N_4_ material (Figure 6a). All samples display an absorption edge at about of 430 nm in wavelength, corresponding to an energy bandgap of 2.7 eV, which can be estimated by using Wood-Tauc plot on the graph of (αhυ)^1/2^ as a function of photon energy (hυ) as shown in Figure 6b. The difference between samples with different Fe concentrations is mainly at the long tail of absorption spectrum, in the range of 430-600 nm (the two-way arrow in Figure 6a). The absorbance of this tail increases gradually with increasing Fe content which can be explained appropriately due to the incorporation of Fe into the g-C_3_N_4_ lattice, resulting in the formation of impurity energy levels in the band gap. Therefore, electrons can absorb the photon and transfer between impurity energy level and conduction/valence band. In addition, Figure 6a also shows the slight red shift of absorption edge, resulting in a reduction of the calculated band gap from 2.7 eV for pure g-C_3_N_4_ to 2.63 eV for FeCN3 and 2.57 eV for FeCN5 samples. 

In order to investigate the influence of Fe content on the separation behavior of electron-hole pairs in Fe-doped g-C_3_N_4_ nanosheets, PL spectra were carried out at room temperature using an excited wavelength at 350 nm (Figure 7a). All samples exhibit photoluminescence emission in the range of 400–600 nm which is consistent with the 430 nm absorption edge. It is obvious that PL intensity of Fe-doped gC_3_N_4_ sample decreases significantly compared to that of pure g-C_3_N_4_ nanosheets. Since the photoluminescence intensity reflects recombination rate of electron–hole pairs, the lower the PL intensity is, the higher the recombination rate is, the sharp decrease in PL intensity as observed indirectly indicates the high separation efficiency of electron–hole pairs which is essential for improving photocatalytic performance. The cause of the reduction in electron–hole pair recombination rate can be attributed to the presence of Fe^2+^/Fe^3+^ ions in g-C_3_N_4_ crystal lattice which acts as electron capture centers. As it is excited, electron receives energy from a photon and transfers from the top of the valence band to the bottom of the conduction band to become a free electron which then easily hop to impurity level of Fe^3+^ due to its position at the middle of the bandgap. As a result, the lifetime of electron-hole pair increases that benefits for photocatalytic performance. Gaussian fitting has been carried out to identify the constituent PL peaks, which shows four major PL centers, including P1 (428 nm, 2.90 eV), P2 (451 nm, 2.75 eV), P3 (483 nm, 2.57 eV), P4 (533 nm, 2.33 eV) (Figure 7b), in accordance with those of previous g-C_3_N_4_ synthesized in Ar atmosphere [44]. As the Fe-doping content increases, the PL band extents slightly towards longer wavelength as shown by horizontal arrow in Figure 7b (Table 2). For example, peak P4 shifts from 533 nm to 534 nm, 436 nm, 540 and 542 nm for g-C_3_N_4_, FeCN3, FeCN5, FeCN7 and FeCN10 samples, respectively, which agrees well with the red shift of absorption edge in Figure 6a. This observation once again identifies certain influence of Fe impurity on the crystal structure and hence optical properties of g-C_3_N_4_ nanosheets.

### 3.4. Chemical Composition Analysis

Chemical composition of Fe-doped g-C_3_N_4_ nanosheets and their electronic energy states were determined using X-ray photoelectron spectroscopy (XPS). Figure 8a presents the survey scan XPS spectra of pure g-C_3_N_4_ and FeCN7 samples which shows that g-C_3_N_4_ nanosheet exhibits characteristic peaks of C, N and O respectively at 284 eV, 397 eV and 532 eV while FeCN7 photocatalyst displays one more sharp photoelectron peak at about 710 eV. The high resolution of C1s XPS spectrum is shown in Figure 8b. C1s XPS spectrum of g-C_3_N_4_ nanosheet contains four sub-peaks at binding energy of 284.6 eV, 286.2 eV, 287.1 eV and 288.3 eV which are assigned to C=N sp^2^, C–N sp^3^, C–O and C=O bonding, respectively [45,46]. These XPS peaks shift to lower binding energy as embedding Fe^3+^ into host crystal (as shown by dash vertical lines in Figure 8b). Figure 8c displays high resolution of N1s XPS spectrum of pure g-C_3_N_4_ and FeCN7 which is separated into three sub-peaks at 395.8 eV, 397.2 eV and 399.1 eV. These peaks can be assigned to C1s states in C–N=C, N–(C)_3_ and C–N–H bonding. In contrast to C1s XPS spectrum, N1s XPS peaks slightly shift towards to higher energy binding, indirectly indicating that Fe^3+^ ion binds to N atom rather than C atom as doped into g-C_3_N_4_ (Figure 1). The presence of Fe^3+^ ions on the surface of g-C_3_N_4_ nanosheets is also verified by high resolution XPS spectrum in binding energy region of 700–740 eV. Two main peaks at 709 eV and 722 eV correspond to multiplet splitting of high spin Fe2p_1/2_ and Fe2p_3/2_ which in turn are separated into two sub-peaks at (708.8 eV, 712.6 eV) and (722.1 eV, 728.1 eV) due to the difference in oxidation states of Fe (Fe^2+^ and Fe^3+^) as shown in Figure 8d. The ratio of oxidation state Fe^3+^:Fe^2+^ can be estimated at 62:38 for FeCN7 photocatalyst.

### 3.5. Photocatalytic Activity Analysis

Photocatalytic performance of the synthesized Fe-doped g-C_3_N_4_ for degradation of RhB under xenon lamp illumination (illuminance E_v_~2500 lx) is shown in Figure 9a. The results indicate that adsorption-desorption equilibrium can be achieved after less than 10 min of stirring in the dark for all samples. After 30 min of stirring in dark, FeCN7 shows the strongest adsorption capacity, reducing RhB concentration by 43%. This can be explained reasonably by the largest specific surface area of FeCN7 as analyzed in BET result. With the presence of pure g-C_3_N_4_ photocatalyst, the degradation of RhB under xenon light irradiation is only of 60% after 60 min of xenon lamp illumination while the photocatalytic performance of Fe-doped g-C_3_N_4_ is improved obviously. The degradation conversion of RhB by Fe-doped g-C_3_N_4_ photocatalyst increases gradually with Fe-doping content, reaches the highest value for FeCN7 and then decreases with further increase of Fe content. Almost 100% of RhB is decomposed after 30 min of xenon lamp exposure for FeCN7 sample. The pseudo-first-order kinetic model is used to determine photocatalytic reaction rate, ln(C_o_/C) = kt, where the rate constant k can be achieved from the slope of the linear relationship of the plot ln(C_o_/C) versus reaction time (Figure 9b). The catalyst FeCN7 exhibits the largest reaction rate constant (k~0.117), which is about ten times greater than that of pure g-C_3_N_4_ (k~0.012). To demonstrate the ability to use sunlight during photocatalytic processes of Fe-doped g-C_3_N_4_ nanosheets, photocatalytic performance under sunlight illumination (illuminance E_v_~7500 lx) was carried out and presented in Figure 9c,d. This also reveals the strongest photocatalytic performance of FeCN7 sample with reaction rate k~0.140, thirteen times larger than that of pure g-C_3_N_4_ nanosheet. This suggests the potential of g-C_3_N_4_-modified photocatalysts in green technique, utilizing and converting solar energy into other forms.

In order to check the reusability of as-prepared Fe-doped g-C_3_N_4_ photocatalyst, the recycling test under xenon lamp illumination was carried out for FeCN7 sample as shown in Figure 9e. It is obvious that as-synthesized Fe-doped well remained photocatalytic performance or possesses good stability, the photodegradation percentage of RhB is >95% after three cycles. Figure 9f shows the temporal evolution of RhB absorption spectrum under xenon lamp irradiation for FeCN7 sample. Besides the significant decrease in the absorbance of RhB over time, the blue shift of the wavelength of absorption maximum was also observed as illuminated by a xenon lamp, which reveals the deethylation of RhB, occurring on the surface of the photocatalyst. It is well known that the blue shift is due to the absorption maxima of intermediate products of RhB deethylation process—RhB (554 nm); N,N,N′-Triethyl-rhodamine (539 nm); N,N′-Diethyl-rhodamine (522 nm); N-Ethyl-rhodamine (510 nm) and finally rhodamine (497 nm) [47,48]. In this case, the blue shift together with the sharp decrease of absorption maxima indicates that there are two photodegradation pathways for RhB coexist and compete—(i) cleavage of the whole conjugated chromophore structure and (ii) N-deethylation.

It is well known that the highest occupied molecular orbital (HOMO) is distributed on N_2_ atoms and the lowest unoccupied molecular orbital (LUMO) is located at C and N_2_ atoms [28,49], photo-generated electrons are difficult to transfer between different heptazine units in g-C_3_N_4_ crystal lattice, resulting to a high recombination rate of electron-hole pairs and limited photocatalytic efficiency. Fe-doping into crystal lattice of g-C_3_N_4_ in interstitial configuration created Fe^3+^ electron-trapped centers which keep important role in photocatalytic activity. Therefore, the mechanism of enhancing RhB decomposition can be explained by the contribution of electron transfer processes (Figure 1) as follows—(i) An electron in valence band absorbs a photon to transfer to conduction band, becoming a free electron and leaving positively charged hole in the valence band. Both electron and hole are free charge carriers and can take part in redox reaction to decompose organic chemical. (ii) Highly oxidized Fe^3+^ ion traps photogenerated electron from N_2_ and C atoms, leading to the longer lifetime of positive charge holes which can oxidize OH^−^ into OH^•^. (iii) After trapping electron, Fe^3+^ ion becomes Fe^2+^ itself can reduce O_2_ into O_2_^•−^ and transform to Fe^3+^. The consequence of this is a low recombination rate of photogenerated electron-hole pairs as demonstrated by PL results. (iv) A part of electrons on Fe^2+^ energy states easily absorb photon in visible range to transfer to conduction band which can reduce O_2_ into O_2_^•−^ and leaving highly oxidized Fe^3+^ ion (Figure 1). This is manifested in the absorption spectra where absorbance increases gradually in the range of 430–600 nm when Fe concentration increases. The enhancement in photocatalytic activity is also supported by the BET measurement which indicates that the specific surface area and pore volume is largest for FeCN7 photocatalyst.

## 4. Conclusions

Fe impurity has been successfully doped into crystal lattice of g-C_3_N_4_ nanosheets in interstitial doping configuration. The incorporation of Fe ions into g-C_3_N_4_ resulted in a slight expansion of crystal lattice, a reduction in energy band gap, an increase in specific surface area and a decrease in photoluminescence intensity. In addition, the analysis showed that Fe impurity existed in g-C_3_N_4_ nanosheets under two different oxidation states Fe^2+^ and/or Fe^3+^ which can be considered as the main causes leading to the reduction of photoluminescence and the enhancement of photocatalytic activity. Fe-doped g-C_3_N_4_ nanosheets exhibited superior photocatalytic performance compared to pure g-C_3_N_4_. In particular, g-C_3_N_4_ nanosheet doped with 7 mol% Fe exhibited strongest photocatalytic activity with reaction rate 10 times higher than that of pure sample, decomposed almost 100% RhB 10ppm solution after 30 min of xenon lamp illumination. A similar result was achieved when decomposing RhB using Fe-doped g-C_3_N_4_ photocatalyst under sunlight, FeCN7 photocatalyst decomposed almost 100% RhB after 30 min, 14 times faster than pure sample, indicating the potential of g-C_3_N_4_ modified photocatalyst in energy conversion field.

## Figures and Tables

**Figure 1 polymers-12-01963-f001:**
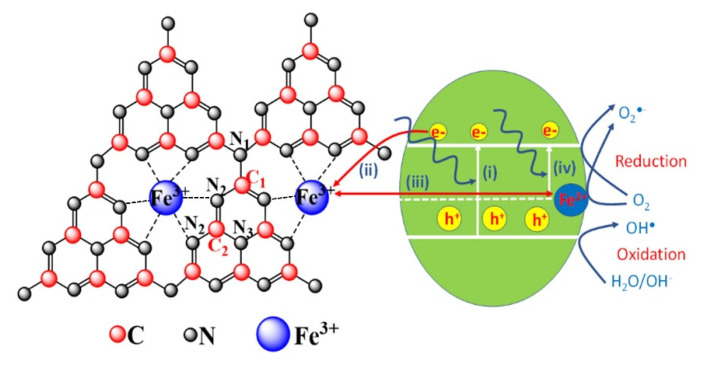
Charge transfer processes in photocatalytic performance of Fe-doped g-C_3_N_4_.

**Figure 2 polymers-12-01963-f002:**
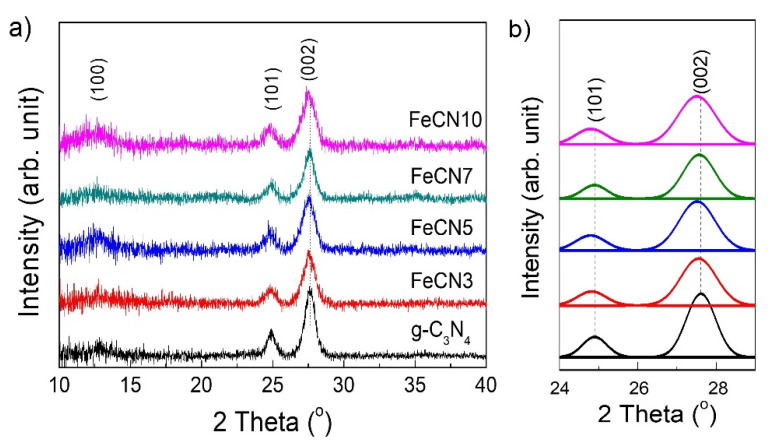
(**a**) X-ray diffraction (XRD) patterns of Fe-doped g-C_3_N_4_ nanosheets with different Fe concentrations; (**b**) The comparison of (101) and (002) XRD peak position.

**Figure 3 polymers-12-01963-f003:**
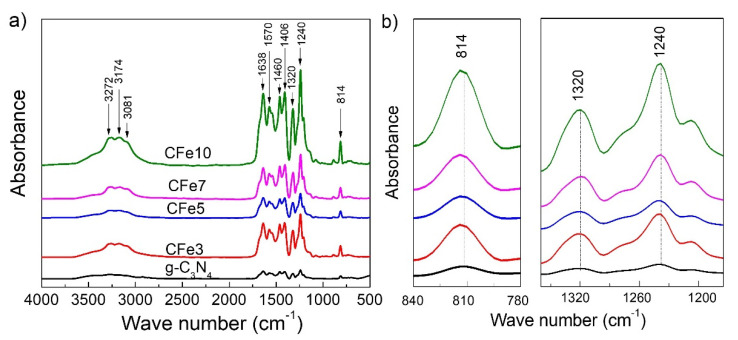
(**a**) Fourier-transform infrared (FT-IR) spectra of Fe-doped g-C_3_N_4_ nanosheets with different Fe concentrations and (**b**) The shift of FTIR peak position.

**Figure 4 polymers-12-01963-f004:**
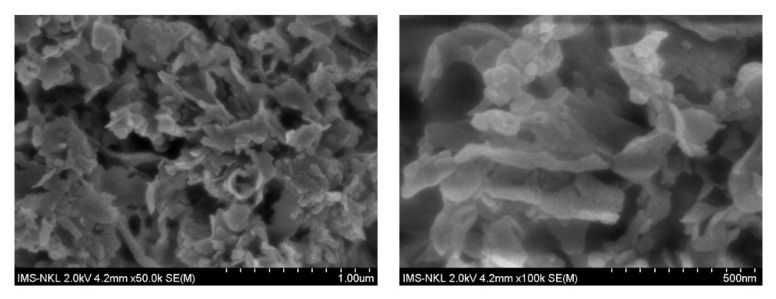
Field emission scanning electron microscopy (FE-SEM) images of pure g-C_3_N_4_ nanosheets with different magnifications.

**Figure 5 polymers-12-01963-f005:**
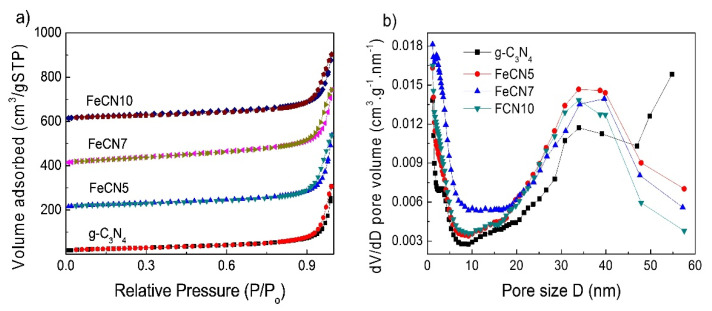
(**a**) N_2_ adsorption-desorption isotherms of Fe-doped g-C_3_N_4_ nanosheets with different Fe concentrations and (**b**) Barrett-Joyner-Halenda (BJH) pore size distribution plots.

**Figure 6 polymers-12-01963-f006:**
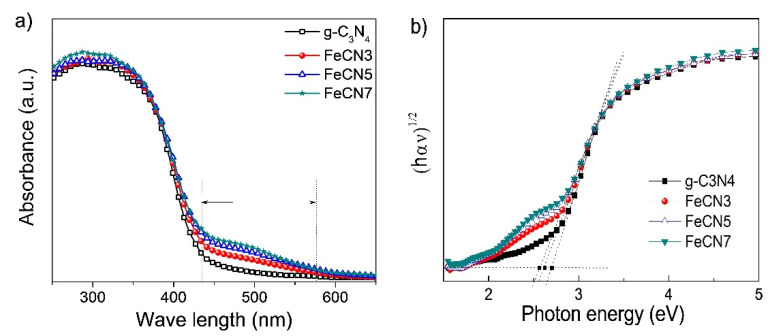
(**a**) Absorption spectra of Fe-doped g-C_3_N_4_ nanosheets with different Fe concentrations and (**b**) Wood-Tauc method in determining the band gap energy of indirect semiconductor.

**Figure 7 polymers-12-01963-f007:**
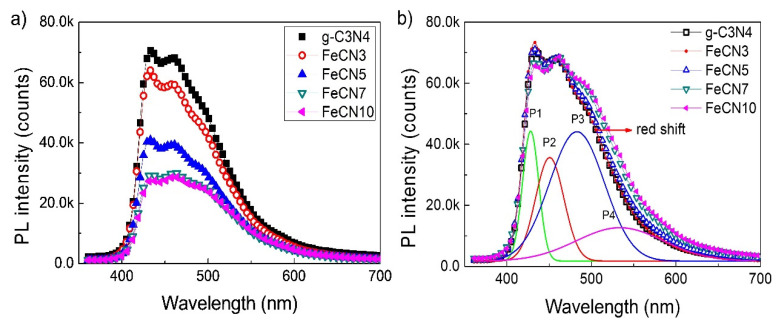
(**a**) Photoluminescence (PL) spectra of Fe-doped g-C_3_N_4_ nanosheets with different Fe concentrations and (**b**) The normalized PL spectra and the Gaussian fitting for pure g-C_3_N_4_.

**Figure 8 polymers-12-01963-f008:**
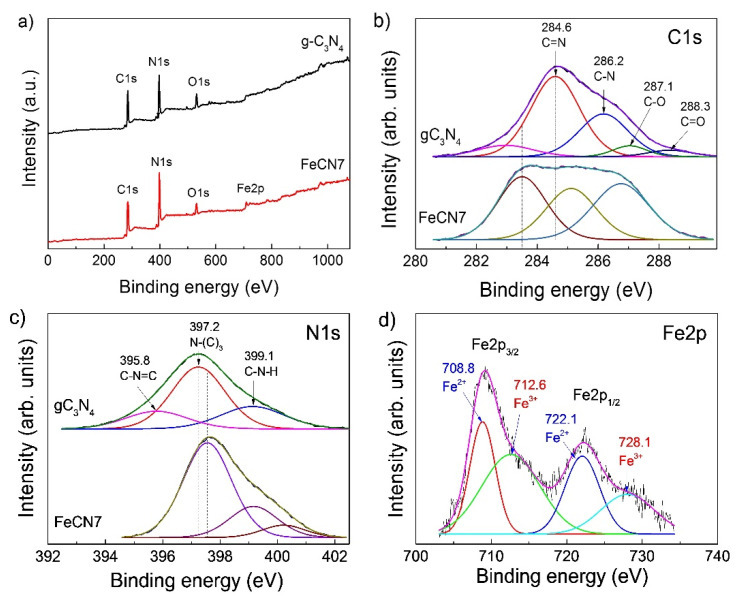
(**a**) X-ray photoelectron (XPS) survey spectra and (**b**) XPS spectra of C1s, (**c**) N1s and (***d***) Fe2p of pure g-C_3_N_4_ and FeCN7 photocatalyst.

**Figure 9 polymers-12-01963-f009:**
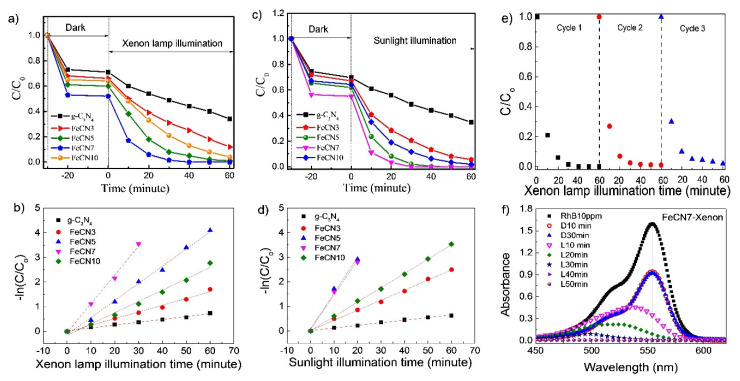
Photocatalytic activities and reaction rate of Fe-doped g-C_3_N_4_ nanosheets with different Fe concentrations in decomposing RhB solution under (**a**,**b**) Xenon lamp illumination and (**c**,**d**) sunlight illumination; (**e**) the reusability of as-prepared Fe-doped g-C_3_N_4_ photocatalyst and (**f**) the change of absorption maxima and absorption intensity of RhB solution at different times.

**Table 1 polymers-12-01963-t001:** The Data of Brunauer–Emmett–Teller (BET) Surface Area, Pore Volume and Pore Size of Fe-Doped g-C_3_N_4_ Nanosheets with Different Fe Concentrations.

Sample	g-C_3_N_4_	FeCN5	FeCN7	FeCN10
**BET surface area** (m^2^/g)	91	100	132	104
**Pore volume** (cm^3^/g)	0.475	0.5223	0.532	0.465
**Pore size** (nm)	21	21	16	18

**Table 2 polymers-12-01963-t002:** The Position of Photoluminescence Peaks P1, P2, P3, P4 of Fe-Doped g-C_3_N_4_ Nanosheets with Different Fe Concentrations.

Sample	g-C_3_N_4_	FeCN5	FeCN5	FeCN7	FeCN10
**Peak 1** (nm)	431.73	429.72	429.15	429.2	429.11
**Peak 2** (nm)	453.85	452.24	451.93	451.75	451.63
**Peak 3** (nm)	488.88	488.27	487.77	489.11	488.48
**Peak 4** (nm)	538.17	539.56	539.15	540.44	539.31

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
