# Peer review of "Fe-Doped g-C3N4: High-Performance Photocatalysts in Rhodamine B Decomposition"

_polymers, 2020, doi:10.3390/polym12091963_

Round 1

Reviewer 1 Report

The work entitled “Fe-doped g-C3N4: high-performance photocatalysts in Rhodamine B decomposition” is a good candidate for publication in Polymers, but I have some comments to be taken seriously by the authors in revising (MAJOR REVISION) the current version of their manuscript:

  • The Figures should be in the order indicated in the text. Referring to Fig.9 in line 75 is not common. It should be modified.
  • The necessity behind doing this research is missing in the introduction. According to the literature that the authors mentioned, Fe-C3N4 was synthesized previously and used as photocatalysts in Rhodamine B decomposition. Which gap in the literature necessitated doing this work?
  • As the authors used Fig. 9 in line 150, I recommend represent Fig.9 as Fig. 1.
  • The FTIR peaks should be redrawn from 4000 to 400 Cm-1.
  • The newly observed band due to Fe doping should be described according to the FTIR spectra.
  • The description of Fig.6 is for Fig. 7 and visa versa please modified it in the text.
  • Conclusion can be modified by more quantitative data.

Author Response

Response to Reviewer 1 Comments

Dear reviewer,

According to your comments and suggestions, we have revised the manuscript as yellowed parts.

  • The Figures should be in the order indicated in the text. Referring to Fig.9 in line 75 is not common. It should be modified.

The Figures have been rearranged in the order of appearance. For example, Figure 9 changed to Figure 1.

  • The necessity behind doing this research is missing in the introduction. According to the literature that the authors mentioned, Fe-C3N4 was synthesized previously and used as photocatalysts in Rhodamine B decomposition. Which gap in the literature necessitated doing this work?

We have added the necessity of the research in the introduction, in Lines 87-90: “Evidence of the improvement of photocatalytic efficiency as well as its mechanism in Fe-doped g-C3N4 material remains to be confirmed, contributing greatly to the discovery of high-performance photocatalyst in particular and solar energy conversion material in general”.

  • The FTIR peaks should be redrawn from 4000 to 400 cm-1.

FTIR peaks have been redrawn from 4000 to 400 cm-1 as shown in Figure 3.

  • The newly observed band due to Fe doping should be described according to the FTIR spectra.

No new FTIR band was observed after doping Fe. The biggest change is that FTIR intensity increased significantly.

  • The description of Fig.6 is for Fig. 7 and visa versa please modified it in the text.

We have corrected this mistake in the manuscript.

  • Conclusion can be modified by more quantitative data.

We have revised the conclusion as shown in the manuscript, the yellow highlighted part in the conclusion.

We extremely thank you for your suggestions.

Yours faithfully,

Authors,

Reviewer 2 Report

The work is interesting, but it is not very original. The contribution is mainly at the level of the synthesis method. It fails to situate it in relation to the state of the art. In order to improve it, I have a few remarks below.

-Lines 91-92: Could you explain how C3N4 is formed from the Urea or give a reference?

-Line 57: ZnWO4 (4 in subscript)

-Line75: the first figure that appears in the text is Fig. 9, it must become Fig.1.

-Line 92: 550°C, Line 99 90°C, Line 102 80°C (always space between number and unit)

-Line 95: 50ml (space between number and unit)

-Line 137-138: Can you keep the same sample colors for both graphs?

-Line 144: what is the detection limit of conventional XRD?

-Line 145-146: has this phenomenon of 'shift to the left side as Fe concentration' been observed for other metals (Mo, Ag...) in literature works? if so, argue with reference?

-Line 163: you say 'increases to 812.1, 813.4, and 813.9 cm-1 for g-C3N4, FeCN5, and FeCN7, respectively while the peaks' how accurate is the measurement?

-Line 185: 'The BET surface area is 91 m2/g, 100 m2/g, 132 m2/g, and 104 m2/g for g-C3N4, FeCN5, FeCN7, and FeCN10, respectively,..' how to explain that specific surface area increases with the incorporation of Fe into g-C3N4 lattice?

-Line 214: Fig. 6a instead of Fig. 7a! and Line 228 and 230: Fig. 6b instead of 7b!

-Line 242, 246, and 249 and 258: Figure 7 instead of Figure 6!

You also need to standardize or use Fig. or Figure!

Author Response

Response to Reviewer 2 Comments

Dear reviewer,

According to your comments and suggestions, we have revised the manuscript as yellowed parts.

  • Lines 91-92: Could you explain how C3N4 is formed from the Urea or give a reference?

Synthesizing g-C3N4 from Urea precursor is a very familiar method, the added references in Line 97 are [41, 42].

  • Line 57: ZnWO4 (4 in subscript)

We have corrected this mistake in Line 57.

  • Line75: the first figure that appears in the text is Fig. 9, it must become Fig.1.

The Figures have been rearranged in the order of appearance. For example, Figure 9 changed to Figure 1.

  • Line 92: 550°C, Line 99 90°C, Line 102 80°C (always space between number and unit)

We have corrected these mistakes in Lines 97, 104, 107.

  • Line 95: 50ml (space between number and unit)

We have corrected this mistake in Line 100.

  • Line 137-138: Can you keep the same sample colors for both graphs?

Two graphs have been changed for similar color as shown in Fig.2. (Lines 141-143)

  • Line 144: what is the detection limit of conventional XRD?

Conventional XRD can usually detect nanoparticles from 2-2.5 nm.

  • Line 145-146: has this phenomenon of 'shift to the left side as Fe concentration' been observed for other metals (Mo, Ag...) in literature works? if so, argue with reference?

The left shift hasn’t been observed in the literature works. The right shift (0.2⁰) was observed in 2104-Fe-doped.

  • Line 163: you say 'increases to 812.1, 813.4, and 813.9 cm-1 for g-C3N4, FeCN5, and FeCN7, respectively while the peaks' how accurate is the measurement?

Although the measurement step is larger than 1 cm-1, values of FTIR peaks are acceptable because they are the result of fitting plot using origin software.

  • Line 185: 'The BET surface area is 91 m2/g, 100 m2/g, 132 m2/g, and 104 m2/g for g-C3N4, FeCN5, FeCN7, and FeCN10, respectively,..' how to explain that specific surface area increases with the incorporation of Fe into g-C3N4 lattice?

It is difficult to explain why. According to our knowledge, Fe atoms fill in gap space created by six lone-pair electron nitrogen atoms on the surface of the g-C3N4 nanosheet. In principle, this can also result in a decrease in N2 adsorption capacity because N2 molecule can insert into this space. However, BET results showed that Fe-doping caused some changes on the surface of g-C3N4 which is benefit for photocatalytic performance.

  • Line 214: Fig. 6a instead of Fig. 7a! and Line 228 and 230: Fig. 6b instead of 7b! Line 242, 246, and 249 and 258: Figure 7 instead of Figure 6!

We have corrected this mistake.

  • You also need to standardize or use Fig. or Figure!

We have standardized the use of Fig. instead of Figure for all positions.

We extremely thank you for your suggestions.

Yours faithfully,

Authors,

Round 2

Reviewer 1 Report

This is a good work and the data presented are sufficient. Authors need to make discussions fulfilled. Moreover, the necessity of this work should be better highlighted in the abstract. I have no concern about this work, as I found it complete.  

Author Response

Response to Reviewer 1 Comments

Dear reviewer,

According to your comments and suggestions, we would like to answer as follows:

- The authors need to make discussions fulfilled.

We have tried our best to discuss issues around all measurements. In particular, the results relate to the photocatalytic enhancement mechanism.

- The necessity of this work should be better highlighted in the abstract.

The nesecessity of this work is to find a suitable explaination for the enhancement of photocatalytic performance of Fe-doped g-C3N4 nanosheet through experimental measurements. Therefore, in the anstract, we would like to present the results of measurements which relate to the enhancement mechanism. Finally, the main mechanism of photocatalytic enhancement is reported: ”The mechanism of photocatalytic enhancement is mainly explained through the charge transfer processes related to Fe2+/Fe3+ impurity in g-C3N4 crystal lattice.” which will be displayed in the disscusion part.

We extremely thank you for your suggestions.

Yours faithfully,

Authors,
